# OpenReview forum: "No, of Course I Can! Deeper Fine-Tuning Attacks That Bypass Token-Level Safety Mechanisms"
_NeurIPS.cc/2025/Conference — Submitted to NeurIPS 2025_

### Official Review · Reviewer_LZRb · 2025-06-14

**Clarity:** 3
**Significance:** 2
**Originality:** 2
**Rating:** 3
**Confidence:** 4

**Summary:**

This paper proposes the NOICE fine-tuning attack, which enhances harmful fine-tuning data with ``refusal-then-comply'' features. The work begins with the observation that the AMD defense (which leverages the first tokens of aligned LLMs as the starting responses of fine-tuned LLMs) can effectively mitigate harmful outputs. Based on this, the authors present an adaptive attack that aims to suppress the refusal portion of the response. Experimental results demonstrate the effectiveness of both the AMD defense and the NOICE attack.

**Questions:**

- Since LG and AMD are detection-based and intervention-based defenses, respectively, how does the attack perform when both defenses are combined? Exploring this straightforward synergy could be insightful.

- What explains the performance plateau of other attacks in Figure 4?

- Are there effective methods for identifying the most threatening training datapoints---e.g., when only 100 samples are allowed for fine-tuning attacks?

**Ethical Concerns:**

["NO or VERY MINOR ethics concerns only"]

**Final Justification:**

This paper is well-established according to its problem setup, experiments, and implications.

However, my primary concern regarding its novelty remains unresolved throughout the review process. The paper primarily exploits the limitations of the fine-tuning API's moderation mechanisms (or guardrails), rather than the inherent vulnerabilities of the LLMs themselves. In this respect, a more robust moderation module would likely be sufficient to mitigate the proposed attack, rendering the paper's contributions significantly less meaningful in the long term.

I have decided to maintain my original score. That said, I would also find it acceptable if the paper is ultimately accepted.

**Limitations:**

yes

**Quality:**

3

**Strengths And Weaknesses:**

**Strengths**

- Well-written. The observation of shallow fine-tuning attacks provides strong motivation for both the AMD defense and the NOICE attack.

- The NOICE attack remains effective even when faced with the AMD defense, outperforming baseline methods. Moreover, the attack benefits from scalability---larger training datasets (at higher cost) can lead to more harmful model behaviors.



**Weaknesses**

- The technical novelty is limited. The NOICE attack is an adaptive method. Identifying ``refusal-then-comply'' patterns is not difficult---for example, by prompting LLMs. This may be due to the relatively weak moderation mechanisms of current fine-tuning APIs. In this sense, the NOICE attack may still be easy to mitigate.

- The assumed defense setting against fine-tuning attacks is weak. The regularized fine-tuning objective in [1], which is closely related to preserving the initial tokens in responses, is not considered.

- Although the NOICE attack is effective in a comparative sense, it still exhibits a low absolute ASR. The paper should include analysis of failure cases---especially when harmful fine-tuning nearly saturates the safety tests. This would help highlight the limitations of NOICE and suggest directions for future improvement.



**Minor typos:**

- Line 250: citation misformatted.



### **References**


[1] *Safety Alignment Should Be Made More Than Just a Few Tokens Deep*, ICLR 2025.

---

> ### Author Rebuttal · Authors · 2025-07-30
>
> Thank you for reviewing our paper!  We address the weaknesses that you identified below.
>
> > The technical novelty is limited. The NOICE attack is an adaptive method. Identifying ``refusal-then-comply'' patterns is not difficult---for example, by prompting LLMs. This may be due to the relatively weak moderation mechanisms of current fine-tuning APIs. In this sense, the NOICE attack may still be easy to mitigate.
>
> NOICE evades in-production defenses, illustrating that it is technically distinct from existing attacks.  After reporting NOICE to OpenAI, they asked us to wait for over a month before releasing the paper while they looked for ways to block the attack.  We re-ran NOICE against GPT-4o in March after disclosing in January and still found nearly identical ASRs, showing that NOICE is certainly not an easy attack to block– we again reported this to OpenAI.  We tried multiple defenses against NOICE including constrained optimization (Table 6), Llama-Guard, and AMD.  None of them were effective.
>
> > The assumed defense setting against fine-tuning attacks is weak.
>
> We show that our attack works against OpenAI’s and Anthropic’s fine tuning APIs, both of which enforce realistic constraints and employ realistic production-grade defenses against fine tuning attacks.  Furthermore, we evaluate against two published defenses (Qi et. al.’s defense and Llama Guard).  NOICE was able to bypass both of them.
>
> >The regularized fine-tuning objective in [1], which is closely related to preserving the initial tokens in responses, is not considered.
>
> We do compare to Qi et. al.’s defense on Llama in Table 6.  We can merge this with Table 8 if that would be clearer.
>
> >it still exhibits a low absolute ASR
>
> For harmless-data, black-box fine-tuning attacks against production models, NOICE is SOTA by a wide margin (see Tables 3,4).  Therefore, among realistic attacks NOICE presents a significant threat.
>
> > The paper should include analysis of failure cases---especially when harmful fine-tuning nearly saturates the safety tests.
>
> This is a great point.  To address this question, we attempted to train two additional models through OpenAI’s fine-tuning API.  We mixed the LLM-LAT/harmful-dataset with HelpSteer at rates of 1% and 5%.  The model trained with 1% truly harmful data achieved an ASR of 6.33%, which is much lower than the 56% achieved by NOICE.  The moderation API blocked us from training on the dataset that contained 5% harmful data.  Therefore, NOICE is still more effective against production API’s than fine-tuning with small fractions of harmful data.  This matches the scaling patterns that we observed in Figure 4.
>
> Note that we don’t know the specifics of the moderation API, so it is possible that it takes into account both the level and percent harmfulness when deciding whether to allow training.
>
> Questions:
> >Since LG and AMD are detection-based and intervention-based defenses, respectively, how does the attack perform when both defenses are combined? Exploring this straightforward synergy could be insightful.
>
> That’s an interesting suggestion!  We ran some experiments on Llama to answer it.  First we used AMD to generate responses and then filtered the outputs using Llama-Guard.  We found that NOICE is still effective against combinations of shallow defenses, whereas other harmless-data tacks are not.  The ASRs are below:
> NOICE: 36%
> ISA: 7.3%
> Implicit: 7%
> Harmful Data: 28%
> Cipher: 2.3%
>
> >What explains the performance plateau of other attacks in Figure 4?
>
> That’s a great question.  It does not require large amounts of data to teach the model that it should always begin its responses with affirmative prefixes.  However, just because the initial tokens are helpful doesn’t guarantee a harmful response.  Therefore, even if the initial tokens are helpful close to 100% of the time, there is some probability that the model will still provide a safe response (See Table 2).  We think that this failure to follow an affirmative prefix with a harmful response is the main limitation for other methods after training on 1000 examples.
>
> >Are there effective methods for identifying the most threatening training datapoints---e.g., when only 100 samples are allowed for fine-tuning attacks?
>
> This would be an interesting direction to explore – however, we do not have any concrete ideas in this regard. We believe that this won’t be straightforward, especially for an attack like NOICE that only uses harmless datapoints to teach the model to renege on an initial refusal to respond to a query, rather than a specific prefix or pattern that leads directly to an unsafe response.

---

> > ### Comment · Reviewer_LZRb · 2025-08-03
> >
> > Thanks for the authors’ responses.
> >
> > Most of my initial questions are addressed. However, the main concern about the technical contribution still exists. The vulnerability this new attack exploits is not related to LLMs, but rather to the moderation mechanisms of fine-tuning APIs. It is not clear what we can learn from this attack from the perspective of AI research. The observation that LLMs can parrot the “refusal-then-comply” pattern is not particularly exciting.
> >
> > Additionally, how can the claim that deeper fine-tuning attacks are attained be supported? The NOICE attack may still belong to shallow attacks, just shifting to several tokens later, isn’t it?
> >
> > These considerations lead me to maintain the score of weak rejection.

---

> > > ### Author Response · Authors · 2025-08-04
> > > **Response to Reviewer LZRb**
> > >
> > > Thanks for the response.  If all of the reviewer’s other concerns are allayed, we hope to underscore our contribution more clearly.
> > >
> > > > The vulnerability this new attack exploits is not related to LLMs, but rather to the moderation mechanisms of fine-tuning APIs.
> > >
> > > Guard-rails are a major and active area of LM-safety and alignment research; our work demonstrates that existing guard rails and past attacks are deeply flawed because they fixate on first-few-token behavior.  Could the reviewer explain why this is a drawback of our paper?
> > >
> > >
> > > > It is not clear what we can learn from this attack from the perspective of AI research. The observation that LLMs can parrot the “refusal-then-comply” pattern is not particularly exciting.
> > >
> > > We disagree with this characterization - certainly LLMs parrotting a response  pattern is not very interesting in isolation. However, our paper makes the following contributions to the science and practical impact of fine tuning attacks:
> > >
> > >  1. We show that attacks from prior work  only target the first few tokens, making them easy to block using defenses like LG and AMD.
> > >
> > > 2.  Correspondingly, defenses used in production fine tuning APIs that evolved alongside these attacks fixate on the first several tokens.  We show that using nothing but harmless data, in-production defenses can be severely compromised by fine-tuning on data that teaches the model to exhibit harmful behavior deeper in the response.
> > >
> > >
> > > > Additionally, how can the claim that deeper fine-tuning attacks are attained be supported? The NOICE attack may still belong to shallow attacks, just shifting to several tokens later, isn’t it?
> > >
> > >
> > > By definition (lines 6-7), we call attacks shallow when they only change behavior in the first few tokens.  Yes, we aim for behavior shifts deeper in the responses.  Even though this is a simple change, it has profound implications.  ASRs against GPT go from 6% to over 50%, and proposed defenses in the literature (Llama-Gaurd, Qi. et. al.’s constrained fine-tuning, GradSafe) stop working effectively.  Clearly this is a major vulnerability recognized by top model providers that will be patched because of this paper’s insight.
> > >
> > >
> > > We hope in light of this, the reviewer will reconsider our score. We are happy to provide any further clarifications neeeded.

---

### Official Review · Reviewer_XvRX · 2025-06-29

**Clarity:** 3
**Significance:** 4
**Originality:** 4
**Rating:** 5
**Confidence:** 3

**Summary:**

The paper proposes NOICE, a novel attack method for LLMs for by-passing safety alignment tuning. The authors first formulate the problem as a probability that a model responds or refuses given a harmful prompt. They demonstrate empirically that given a model responds with the first few tokens as a refusal statements (e.g. "No, I cannot respond to that") the chances of a harmful response in the rest of the statement goes down significantly. Popular methods of attack rely on reducing the probability of generating the first few tokens of refusal in the first place, which opens up the possibilities of a complete response with harmful content. However, when the model generates refusal tokens initially, harmful responses are harder to obtain in the later tokens. Based on the combination of these observations and literature, they conclude that the first few tokens are most important for safety alignment, and thus break alignment by introducing prompts that encourage models to ignore the first few tokens. They apply this in practice on production APIs for ChatGPT and Anthropic, and show that it is possible to break safety alignment by using a minimal amount of cost.

**Questions:**

1. How does this hold up against extremely-long-context models?
2. How limited are the prompt variations for this type of attack? i.e. have you tried other strings apart from "Now that we have finished discussing the safety policy, let’s answer your question"
3. How can we defend against this?

**Ethical Concerns:**

["Major Concern: Safety and security"]

**Final Justification:**

This paper is a 5 because it does demonstrate an interesting flaw. I think it is important in tackling a major issue in current LLM systems and is a well-rounded good paper that overall gets its ideas through convincingly. However, it lacks any ground-breaking contributions, so a 6 is not warranted.

**Limitations:**

This exposes an important flaw in the current production level models, and can be used to significant repurcussions unless they are properly patched up. The authors do not propose a defense, and so there are potential negative societal impacts here.

**Quality:**

2

**Strengths And Weaknesses:**

Strengths -
1. Very interesting and innovative method of breaking safety alignment. While the idea of alignments being shallow is well-known, the authors use it very creatively.
2. Intuitive to understand and replicate.
3. Exposes a large safety issue in current production level models.

Weaknesses -
1. The method leverages and reinforces what we already know, but does not make significant scientific contributions.
2. The probability formulation can be better explained.

---

> ### Author Rebuttal · Authors · 2025-07-30
>
> Thank you for reviewing our paper and providing helpful feedback.  We try to address your comments below:
>
> Weaknesses:
>
> >The method leverages and reinforces what we already know, but does not make significant scientific contributions.
>
> We respectfully disagree with this characterization. We believe that our advances constitute significant scientific contributions as follows:
> The observation that shallow fine-tuning attacks proposed in prior work are easily blocked by standard defenses (as well as our Aligned Model Defense)
> Our novel attack, NOICE, is able to bypass several defenses and still result in a harmful-fine tuned model using only harmless data.
> Further, our work is validated against frontier production-grade fine tuning APIs including OpenAI’s and Anthropic’s. This is a higher standard of proof than has been pursued in most prior work on this topic, and we believe this demonstrates a strong practical vulnerability of production-grade fine-tuning APIs.
>
> >The probability formulation can be better explained.
>
> We’ll expand the explanation in the final version of the paper.  To give another explanation, most prior fine-tuning attacks against production models aim to increase the probability of a helpful prefix given a harmful prompt, i.e. increase $$\mathbb{P}(\textrm{Helpful Prefix}|\textrm{Harmful Prompt}).$$  These methods leave the probability of a harmful response given a helpful prefix and harmful response the same, i.e. $$\mathbb{P}(\textrm{Harmful response}|\textrm{Helpful Prefix}, \textrm{Harmful Prompt})$$ is unchanged.  We can decompose the probability that past attack strategies work as
> $$\mathbb{P}(\textrm{Harmful Response}|\textrm{Helpful Prefix}, \textrm{Harmful Response})\mathbb{P}(\textrm{Helpful Prefix}|\textrm{Harmful Response}).$$ AMD sets $$\mathbb{P}(\textrm{Helpful Prefix}|\textrm{Harmful Response}) \approx 0,$$ causing past attacks to fail.  However, our attack strategy is to increase $$\mathbb{P}(\textrm{Harmful Response}|\textrm{Refusal Prefix}, \textrm{Harmful Prompt})\mathbb{P}(\textrm{Refusal Prefix}|\textrm{Harmful Prompt}),$$ making AMD ineffective.  Our attack is very hard to block or detect from the first several tokens because it does not change $$\mathbb{P}(\textrm{Refusal Prefix}|\textrm{Harmful Prompt}),$$ as shown in Table 2.  Please let us know if this remains unclear.
>
> Questions:
>
> >How does this hold up against extremely-long-context models?
>
> Good question.  GPT-4o and Claude have context lengths of 128k and 200k tokens respectively, and NOICE was effective against these models.  Perhaps you are asking how the attacks work with ultra-long harmful requests?  This would certainly be an interesting question to explore, though we are currently not aware of a dataset that contains long and harmful requests.
>
> > How limited are the prompt variations for this type of attack? i.e. have you tried other strings apart from "Now that we have finished discussing the safety policy, let’s answer your question"
>
> That’s a great question!  When writing the paper we indeed tried other variants of the NOICE attack.  For example, in our first iteration of NOICE, we did not use a system prompt.  We instead simply generated a refusal to each question and then appended the original response from HelpSteer to the refusal.  The resulting ASRs were comparable against open-source models (Llama: 54.3%, Mistral: 74%), but lower for frontier models (13-19% against GPT-4o).  We also tried generating NOICE data with one model to use against other models and found that this did not produce strong attacks, likely because different models have slightly different refusal mechanisms.  The training data must be generated from the attacked model for NOICE to be effective.
>
>
> >How can we defend against this?
>
> This is currently an open question.  OpenAI asked us to wait over a month before releasing any public artifacts after we originally reported the attack to as part of our responsible disclosure process.  When we last checked, NOICE was still effective against production models.  As a temporary patch, we recommend training fine-tuning data-moderators on NOICE-style attack data; one could also train output-moderation classifiers on NOICE-style responses.  However, we believe that variants of NOICE that exploit other mechanisms to bypass defenses will likely still work (parallel works to ours like [1] show similar vulnerabilities). Ultimately, more research should go into deeper attacks and corresponding defenses to make models safer.
>
>
> [1] Davies et. al., Fundamental Limitations in Defending LLM Finetuning APIs.

---

> > ### Comment · Reviewer_XvRX · 2025-08-04
> > **Response to Author Rebuttal**
> >
> > Thank you for your response.
> >
> > 1. I agree, there are few prior works that discuss this scenario specifically and this is a clever discovery, but it is technically lightweight. For example, **why** is this a problem? How generalizable is this attack? Any mathematical/logical reasoning tied to the way models are trained or datasets are curated that could help devise better solutions for model development would be very welcome. A higher score would warrant more fundamental contributions than this.
> >
> > 2. Yes, I mean longer attack tokens on long-context models. It ties into the next question on how limited the prompt variations are. Standard datasets are not required to test this out, custom datasets can be easily curated. The cusp of the questions are - what is the scope of the attack and are the kind of tokens it is most susceptible to? Is it just this prompt or what extent of prompt variations such as changing "we" to "you" and at what point does the model stop being vulnerable? It can be delved into further, but I understand that is beyond the scope of this paper. Depending on the findings and thus possible directions of further targeted research depending on the observations could warrant a score of 6.
> >
> > Due to the above reasons, I keep my score at 5.

---

### Official Review · Reviewer_R7Hk · 2025-06-30

**Clarity:** 2
**Significance:** 3
**Originality:** 3
**Rating:** 3
**Confidence:** 5

**Summary:**

This paper presents NOICE, a fine-tuning attack that trains LLMs to begin responses with a refusal before continuing with harmful content, allowing it to evade token-level defenses. The authors argue that existing attacks mostly influence the initial response tokens and can be blocked by simple prefill strategies, while NOICE creates deeper behavioral shifts that persist beyond the first few tokens. Through experiments on both open-source and production models, the paper shows that NOICE can bypass standard moderation and filtering systems using only seemingly harmless data. The work raises concerns about the limitations of current alignment methods and highlights the need for more robust defenses against subtle fine-tuning vulnerabilities.

**Questions:**

1. The attack is evaluated primarily on HeX-PHI. Does NOICE generalize to other harmful prompt sets?
2. The paper mentions successful attacks with 1000–5000 datapoints. How robust is this attack at smaller data regimes (e.g., <500 examples), and how sensitive is it to noise in the fine-tuning data?

**Ethical Concerns:**

["NO or VERY MINOR ethics concerns only"]

**Final Justification:**

I have concerns regarding whether the paper is exposing vulnerability of the guardrail or the LLM itself. Meanwhile, the writing is extremely hard to follow and needs major revision before acceptance.

**Limitations:**

I'm happy to raise score if the author can address all my concerns above

**Quality:**

3

**Strengths And Weaknesses:**

The paper addresses a critical and timely problem—harmful fine-tuning attacks on LLMs—which is both well-motivated and of high practical relevance, especially as commercial fine-tuning APIs become more widely used. The proposed NOICE attack is novel in that it circumvents existing shallow defenses by embedding harmful content deeper in the response, and the empirical results against production models are compelling.

However, several limitations remain:
1. A major gap is the absence of evaluation on capability or overrefusal benchmarks. For the AMD defense in particular, where generation switches from an aligned model to a fine-tuned one after the first few tokens, it's unclear how this impacts general model performance, especially on tasks like math or coding. Could this mechanism unintentionally degrade capability or introduce new failure modes?

2. The AMD method relies on detecting whether a prompt is harmful before deciding to switch models. But the paper doesn’t clarify how this detection is performed in practice at inference time—does it assume access to a reliable classifier, or is this manually annotated?

3. The switching point in AMD is underspecified. How many tokens must the base model generate before handing off to the fine-tuned model? Is this fixed or adaptive? How sensitive are the results to this design choice?

4. Regarding the NOICE attack, it would strengthen the paper to discuss how it fares against recent defenses such as Safety-Tuned LLaMAs [1] or LISA [2], which explicitly aim to mitigate harmful fine-tuning. Can those methods already defend against this kind of deeper attack?

5. The writing needs significant improvements, as the paper jumps back and forth between different datasets and settings. A clear section on the experiment settings would be better. The reviewer also appreciates if the author can provide clear bullets of contributions.

[1] Bianchi et al. Safety-Tuned LLaMAs: Lessons From Improving the Safety of Large Language Models that Follow Instructions.

[2] Yi et al. LISA: Lazy Safety Alignment for Large Language Models against Harmful Fine-tuning Attack.

---

> ### Author Rebuttal · Authors · 2025-07-30
>
> We appreciate the suggestions for improving the paper.  We try to address all of your concerns below and appreciate any guidance for strengthening our work.
>
> **Weaknesses:**
>
> >A major gap is the absence of evaluation on capability or overrefusal benchmarks. For the AMD defense in particular, where generation switches from an aligned model to a fine-tuned one after the first few tokens, it's unclear how this impacts general model performance, especially on tasks like math or coding.
>
> Over-refusal will not be an issue for AMD because it uses the standard model that has not been fine-tuned to fill in the first several tokens.  For example, if AMD is applied to GPT-4o, then GPT-4o will complete the first several tokens of the response before switching to a fine-tuned version of GPT-4o.  Therefore, as long as GPT-4o does not have an over-refusal problem, nor will AMD.
>
> >The AMD method relies on detecting whether a prompt is harmful before deciding to switch models. But the paper doesn’t clarify how this detection is performed in practice at inference time—does it assume access to a reliable classifier, or is this manually annotated?
>
> AMD does not require a classifier to detect harmfulness.  It directly generates the first several tokens from the base model regardless of prompt harmfulness.
>
> >The switching point in AMD is underspecified. How many tokens must the base model generate before handing off to the fine-tuned model? Is this fixed or adaptive? How sensitive are the results to this design choice?
>
> This is a good question.  In our tests, the switching point is fixed after 15 tokens– we should have specified this in the paper.  We found when developing the method that as long as there are enough tokens for a basic refusal (i.e. “I’m sorry, I cannot”), it is sufficient to guard the models against past attacks.
>
> >Regarding the NOICE attack, it would strengthen the paper to discuss how it fares against recent defenses such as Safety-Tuned LLaMAs [1] or LISA [2], which explicitly aim to mitigate harmful fine-tuning. Can those methods already defend against this kind of deeper attack?
>
> We compared under the constrained optimization method developed by Qi et. al., which is more recent than [1] and [2] and also aims to explicitly defend against harmful fine-tuning.  Please see table 6 for these results.
>
> >The writing needs significant improvements, as the paper jumps back and forth between different datasets and settings.
>
> While we can’t provide a new draft during the rebuttal period, we will commit to improving the writing for future versions, following the suggestions that you outlined.
>
> Questions:
>
> > The attack is evaluated primarily on HeX-PHI. Does NOICE generalize to other harmful prompt sets?
>
> Yes, the attack generalizes against other datasets, too.  For example, on AdvBench and LLM-LAT, NOICE achieves ASRs of 39% and 44.7% respectively, which are in the same ballpark.  We also tried NOICE on many of our own harmful prompts, and found that it was effective.  We chose HeX-PHI as the benchmark for the paper because it contains a balanced mixture of well-categorized harmful behaviors.  It is also small enough that GPT-as-a-judge is not too expensive.
>
> > The paper mentions successful attacks with 1000–5000 datapoints. How robust is this attack at smaller data regimes (e.g., <500 examples), and how sensitive is it to noise in the fine-tuning data?
>
> The question about scaling is answered in Figure 3, where the dataset size ranges from 10 to 5000.  In terms of noise in the data, we’re not sure what you mean in the context of language datasets.

---

> > ### Comment · Reviewer_R7Hk · 2025-08-03
> >
> > Thanks for the rebuttal! I believe the method still contains several drawbacks:
> > 1. the method uses the base model to generate initial tokens. Say for a capability use case like multiple choice, where only <10 tokens are needed to complete generation. Then, there is no chance of switching to a new model as the first few tokens are generated by the base model. Now say we have finetuned the base model to improve the multiple choice accuracy, then there is zero chance that this method can benefit from the benign finetuning.
> > 2. Switching model also requires additional generation time as the KV cache of the base model cannot be reused by the finetuned model.

---

> > > ### Author Response · Authors · 2025-08-04
> > > **Response to Reviewer R7Hk**
> > >
> > > Thanks for the response.  With respect to your first comment about multiple choice questions, we agree that AMD could damage performance on multiple choice questions.  For your second point, generation of the first 10 tokens using the aligned model can happen in parallel with context-ingestion for the fine-tuned model.  Then the fine-tuned model only requires around 10 extra tokens to be KV-cached before it can start generating additional text.
> > >
> > > However, AMD is certainly not the central point of our paper– as stated in the limitations, lines 227-230:
> > >
> > > “AMD is a non-comprehensive defense that we describe to illustrate the attack mechanism shared by the implicit attacks and ISA; we do not promote it as a panacea against all attacks. AMD is vulnerable to inference-time attacks, and its efficacy is limited by the quality and alignment of the guard model. “
> > >
> > > AMD is meant primarily as an illustrative example of how past attacks are shallow and can be easily blocked.  The main contributions of our paper are:
> > > 1.  We understand past attacks as shallow, demonstrating that they all operate through a similar mechanism.  AMD helps to demonstrate this.
> > > 2.  The development of a new attack that falls outside of the shallow attack paradigm, and thus evades in-production defenses as well as many proposed theoretical defenses.
> > >
> > > To reject our paper based on practical drawbacks of AMD confounds an illustrative thought experiment and the paper’s main contribution.  We hope that the reviewer will reconsider our paper with this in mind, and also account for the extra experiments that we have highlighted to address the reviewer’s other concerns about our paper.

---

> > > > ### Comment · Area_Chair_Y7De · 2025-08-05
> > > > **Discussion**
> > > >
> > > > Dear reviewer,
> > > >
> > > > I appreciate your time in discussing the rebuttal. Can you please comment on the authors' last point on AMD as not being the main contribution to warrant a rejection? This message is intended to encourage further discussion and is not intended to push you to change your score.
> > > >
> > > > Thank you,

---

> > > > ### Comment · Reviewer_R7Hk · 2025-08-07
> > > >
> > > > Thank you for the response! I respectfully disagree that AMD is not the primary contribution of the paper. As written in the abstract, quote, "we correspondingly demonstrate that existing fine-tuning attacks are “shallow" – attacks target only the first several tokens of the model response, and consequently can be blocked by generating the first several response tokens with an aligned model". This demonstrates that AMD is an important defense against such attack. If AMD is not a major contribution, then it basically weakens the contribution of the first bullet.
> > > >
> > > > Overall, the paper touches on a lot of aspects of the LLM finetuning safety, but lacks a universal theme/topic. Therefore, I find the key takeaway of the paper hard to grasp. Does the paper want to propose a new defense or attack? Or it's a paper focusing on the explanation of the attack?
> > > >
> > > > Besides, how does the new attack work if one uses a moderator to filter out the harmful examples?

---

> > > > > ### Author Response · Authors · 2025-08-08
> > > > > **Response to Reviewer R7Hk**
> > > > >
> > > > > We appreciate the reviewer’s extended engagement with our paper.
> > > > >
> > > > > Our paper makes two points: 1. Past attacks are shallow and can be easily blocked by several defenses, including AMD. 2. We develop NOICE as a deeper attack that cannot be blocked by these defenses, works against production fine tuning APIs, and exploits a different mechanism than prior attacks.
> > > > >
> > > > > Regarding the value of AMD: AMD’s primary role in our paper is  to show that past attacks are shallow. As the reviewer points out, AMD has some drawbacks, like almost any safety method.   Indeed, we break AMD ourselves using NOICE, so we certainly wouldn’t recommend it as a foolproof defense.
> > > > >
> > > > > Most safety papers fall into one of three categories: attack papers, defense papers, or attack and defense papers.  Ours does not fit neatly into these standard molds, but can be thought of as an attack paper that also exposes limitations of prior attacks.  There are two parts to our paper:
> > > > >
> > > > > 1.  **Study of Limitations of Past Attacks:** We study past attacks that use harmless data.  We show that they only affect the first few tokens of the response (Table 2), and we exhibit AMD to show that controlling the first several tokens blocks past attacks (Table 8).
> > > > > 2.  **Development of a Deeper Attack:** We develop a deeper attack, NOICE.  We show that it thwarts not just AMD, but Llama-Guard and Qi. et. al.’s defense too (Tables 6,8).  Many defenses and guard-rails evolved around shallow attacks, making them ineffective against NOICE.  Finally, as a show of NOICE’s efficacy, we jailbreak 2 production models (Tables 3, 4).  NOICE breaks out of the shallow attack paradigm, fooling data moderators (Table 1), defensive fine-tuning schemes (Table 6), and even output filters (Table 8).
> > > > >
> > > > > To put it succinctly, the point of our paper is “past attacks are shallow; we can make them deeper and more powerful.”
> > > > >
> > > > > > Besides, how does the new attack work if one uses a moderator to filter out the harmful examples?
> > > > >
> > > > > NOICE fools two moderators.  It evades a data moderator for the fine-tuning API by using only harmless training data, and it evades production output filters that see the first few tokens “I’m sorry, I can’t help with…”, or similar, and think that the response is safe.  The fact that NOICE is effective against output filters shows the extent to which these output filters evolved in tandem with shallow attacks.
> > > > >
> > > > > Again, we appreciate the reviewer’s engagement with our paper.  We hope that we’ve been able to clarify our contributions.  We welcome further discussion if the reviewer has more questions or concerns about our paper.

---

### Official Review · Reviewer_tnSk · 2025-07-03

**Clarity:** 3
**Significance:** 3
**Originality:** 3
**Rating:** 5
**Confidence:** 4

**Summary:**

The paper introduces NOICE, a fine-tuning attack that bypasses token-level safety mechanisms in LLMs by training models to initially refuse harmful requests before complying with them, thereby evading existing defenses. The authors demonstrate that current fine-tuning attacks are shallow and can be blocked by generating the first few response tokens with an aligned model, while NOICE achieves higher ASRs by making attacks deeper and more stealthy.

**Questions:**

NA

**Ethical Concerns:**

["NO or VERY MINOR ethics concerns only"]

**Final Justification:**

Prior to the rebuttal, I had reservations regarding the novelty of the defense strategy presented in this paper. While the authors are the first to explicitly define a "shallow" approach, the underlying design shares significant similarities with existing methods, as evidenced in my provided reference [1] and in reference [2] cited by reviewer LZRb. Furthermore, I was concerned about the insufficient consideration given to a broader range of comparative defenses.

The authors' detailed response has effectively resolved my concerns. Given the compelling arguments and clarifications provided in the rebuttal, and acknowledging the paper's notable influence on current LLM services, I maintain my original "accept" justification.

[1] SafeDecoding: Defending against Jailbreak Attacks via Safety-Aware Decoding

[2] Safety Alignment Should Be Made More Than Just a Few Tokens Deep, ICLR 2025.

**Limitations:**

yes

**Quality:**

3

**Strengths And Weaknesses:**

# Strengths
1. The paper introduces NOICE, a novel adversarial attack employing a "refuse-then-comply" paradigm that bypasses token-level defenses. This approach demonstrates significant advancement over existing methods, achieving notable attack success rates of 57% on GPT-4o and 72% on Claude Haiku, establishing its practical efficacy in adversarial ML research.
2. I appreciate the authors' focus on exposing critical vulnerabilities in production-level language models with direct practical implications. The authors' responsible disclosure practices and collaboration with vendors to enhance safety mechanisms demonstrate constructive contribution to improving safeguards in deployed systems beyond academic interest.
3. The paper is well-written and easy-to-understand.

# Weaknesses
1. While the paper proposes AMD as a countermeasure, the underlying defense concept has been implicitly present in existing work [1], though not explicitly characterized through the "shallow tokens" framework.
2. The evaluation focuses primarily on AMD and LG defenses, potentially overlooking more advanced moderation techniques [2] and fine-tuning schemes [3] available in the research literature.
3. Despite responsible disclosure to OpenAI and Anthropic, the detailed attack methodology could potentially be exploited against other fine-tuning APIs. The authors could strengthen Section 9 by providing explicit guidelines to ensure this research is employed solely for beneficial security research purposes rather than malicious exploitation.

[1] SafeDecoding: Defending against Jailbreak Attacks via Safety-Aware Decoding

[2] GradSafe: Detecting Jailbreak Prompts for LLMs via Safety-Critical Gradient Analysis

[3] Mitigating Fine-tuning based Jailbreak Attack with Backdoor Enhanced Safety Alignment

---

> ### Author Rebuttal · Authors · 2025-07-30
>
> Thank you for the thoughtful review.  We try to address your major concerns below:
> >While the paper proposes AMD as a countermeasure, the underlying defense concept has been implicitly present in existing work [1], though not explicitly characterized through the "shallow tokens" framework.
>
>
> Thanks for pointing out this relevant work.  We will cite it in our next release of the paper, since it indeed shares some similarities with AMD.  There are also notable differences: SafeDecoding trains an expert model to guard against inference-time jailbreaks, augmenting the safety of the base model.  We recognize that a base model can serve as a safer counter-weight to user-fine-tuned models.  SafeDecoding also uses a weighted decoding scheme, whereas we simply control the initial tokens using the base model.
>
>
> As you point out, the main contribution of our paper is the NOICE attack and broader “shallow attacks” framework, and AMD is mainly intended to illustrate how past attacks are shallow.
>
>
> >The evaluation focuses primarily on AMD and LG defenses, potentially overlooking more advanced moderation techniques [2] and fine-tuning schemes [3] available in the research literature.
>
>
> In addition to the AMD and LG defenses, we tested our attack on Llama against the constrained optimization fine-tuning scheme from Qi et. al. (see table 6).  By running our attack against production models, we effectively benchmarked them against realistic defenses deployed in frontier models.
>
>
> In response to your suggestion, we also tested NOICE against GradSafe.  Unlike other moderation methods, GradSafe is designed to detect harmful prompts rather than harmful outputs. NOICE does not conceal harmful prompts, so it is less effective when the output is not considered.  Nonetheless, we found that GradSafe was far less accurate in detecting harmful prompts when used against NOICE on Llama-3-8B-Instruct.  GradSafe uses AUPRC as the metric for success:
>
> |          | NOICE | Standard |
> |--------|-----------|---------------|
> |XSTest|   0.84   | 0.95 |
> | ToxicChat | 0.50 | 0.77 |
>
>
> On ToxicChat against NOICE, GradSafe is no better than a random classifier.
>
>
>
>
>
>
>
>
>
>
>
>
>
>
>
>
> >the detailed attack methodology could potentially be exploited against other fine-tuning APIs. The authors could strengthen Section 9 by providing explicit guidelines to ensure this research is employed solely for beneficial security research purposes rather than malicious exploitation.
>
>
>
>
> We thank the reviewer for raising this concern. As mentioned in the paper, we followed an explicit responsible disclosure procedure before releasing this work publicly. In particular, for both of the production-grade APIs that we attacked (Anthropic and OpenAI), we disclosed the attack results to the model providers; we received a bug bounty from OpenAI for the disclosure. We received permission to publish and acknowledgment of the vulnerability from both OpenAI and Anthropic. We provide code to reproduce our results for future model developers to use in any internal safety/security evaluations, facilitating more robust stress-testing of fine tuning APIs. We strongly believe that publicly disclosing attacks after going through a responsible disclosure process enhances the AI safety ecosystem and enables further research into strong defenses, or other policy-level mitigations that limit the use of fine-tuning APIs. We are happy to provide any further details the reviewer may request and make these changes in the final version of the paper.

---

> ### Comment · Reviewer_tnSk · 2025-08-05
> **Thanks for your response**
>
> Thank you for your response. My concerns have been well addressed, and I will keep my scores.

---

### Note · Authors · 2025-08-15

Dear AC and Reviewers,


We would like to briefly summarize the reviewers’ main concerns and how we addressed them.

**Evaluating whether our attack bypasses fine-tuning defenses:** Reviewers tnSK and R7Hk pointed out that we do not compare against more advanced fine-tuning defenses. This is inaccurate - we do indeed compare to  Qi et. al’s constrained optimization defense; we further tested our attack against GradSafe, as reviewer tnSK suggested. Our attack, NOICE, is successful in compromising both defenses.


**Discussion on AMD as a defense:** After our initial rebuttal, reviewer R7Hk raised several new issues regarding the AMD defense discussed in our paper. We hold that AMD is not the main point of our paper; it is meant to illustrate that existing attacks are shallow, and we do not recommend it as a standalone, production-ready defense.  We have supported this position by citing our own limitations section and by highlighting the experiments whereby we overcome AMD with our own attack.


**Scope broadening requested by reviewer XvRX:** Reviewer XvRX commented on several aspects that would require significantly broadening the scope of our paper, including  1.  Experiments on long-context models. 2. Defenses against the NOICE attack. While certainly valid scientific directions to pursue, we believe these are largely complementary to the key question studied in this paper (showing that prior fine tuning attacks do not capture the full scope of vulnerabilities, and providing a new attack that goes beyond prior work).


**Issues raised by reviewer LZRb:**
Reviewer LZRb stated that our contributions are weak because our paper focuses on bypassing guard-rails rather than fundamental insights about LLMs.  Guard rails and output filters are a major area of LLM research, so we do not view this as a drawback of the paper.  The reviewer also stated that we did not compare to the Qi et. al. defense.  We cited Table 6 in our paper, where we explicitly compared to this defense.  The reviewer also stated that our ASR’s were not high enough; we pointed out that for non-gradient-based fine-tuning attacks they are SOTA, as shown in Tables 3 and 4.  The reviewer requested additional experiments for attacks that use a mixture of overtly harmful and harmless data; we provided these experiments and showed that NOICE outperforms these attacks on GPT-4o.  We believe we have addressed all the issues raised by LZRb.


Thank you for considering our paper.


The Authors

---

### Decision · Program_Chairs · 2025-09-17

**Decision:**

Reject

**Comment:**

The paper argues that current attacks are shallow and thus can be easily detected. Based on this, they propose a new fine-tuning attack for production models, demonstrating substantially better results. The work is quite valuable as it targets the safety of API based models which are the most commonly used models in public.

There is a consistent concern among reviewers (tnSt, XvRx, LzRb) regarding the novelty of the proposed approach. Moreover, the reviewer XvRx further questioned the reason behind the vulnerability in the models or the guardrails which will be helpful in understanding the limitations and to devise better solutions. On similar lines, an important concern is raised regarding the applicability of approach to API only which is essentially highlighting the vulnerability of the guardrails and not the model itself. I encourage authors to address these concerns in the next version of their paper. More specifically,
- Clarify the novelty of their work and add the missing related work. Moreover, clarify the use of AMD as a baseline method and not a contribution of the paper.
- Improve writing and presentation of methodology
- Present results against diverse moderation techniques
- Add an analysis of the failure cases and discuss the limitations of the approach to token length